

# Computational study of potential inhibitors for fat mass and obesity-associated protein from seaweed and plant compounds

Lavanya Prabhakar and Dicky John Davis G

Department of Bioinformatics, Faculty of Engineering and Technology, Sri Ramachandra Institute of Higher Education and Research, Chennai, Tamil Nadu, India

## ABSTRACT

**Background:** Over the past three decades, with substantial changes in lifestyle, the tendency to gain weight has increased, which is resulting in significant consequences affecting an individual's well-being. The fat mass and obesity-associated (FTO) gene is involved in food intake and energy expenditure and plays a crucial role in regulating homeostasis and controlling energy expenditure by hindering signals that generate from the brain. Edible seaweeds have been shown to enhance satiety owing to their health benefits.

**Methods:** Extensive screening of plant-derived anti-obesity compounds and seaweed compounds was conducted and validated for ADME properties and toxicity prediction. Further, the top ranked compounds were docked against the FTO protein to identify potential inhibitors and were subjected to molecular dynamic simulation studies to understand the binding stability of ligand protein complex. Finally, MM/PBSA studies were performed to calculate the binding free energy of the protein-ligand complexes.

**Results:** Through the virtual screening of 1,210 compounds, 443 compounds showed good docking scores less than −7.00 kcal/mol. Drug likeness screenings of 443 compounds showed that only 369 compounds were in accordance with these properties. Further toxicity prediction resulted in 30 non-toxic compounds. Molecular docking studies revealed four top ranked marine compounds. Finally, RL074 (2-hydroxyluzofuranone B) and RL442 (10-acetoxyangasiol) from marine red alga *Laurencia sp* showed good stability from molecular dynamic simulation studies. MM/PBSA results revealed that BT012 (24ε-hydroperoxy-6β-hydroxy-24-ethylcholesta-4,-28(29)-dien-3-one), an oxygenated fucosterol from brown alga *Turbinaria conoides*, possessed higher binding energy. Hence, with all the data obtained it could be concluded that three seaweed compounds, BT012, RL074 and RL442, may act as a potential anti-obesity lead compound in targeting FTO.

Corresponding author
Dicky John Davis G,
dicky@sriramachandra.edu.in

## INTRODUCTION

Globally, the prevalence of obesity and overweight has extensively increased four-multiples from 4% to 18% across different age groups including adults and children from 1975 to 2016 (WHO). Obesity is mainly characterized by the excess accumulation of body fat (*Torres-Fuentes et al., 2013*). In modern times, owing to a sedentary lifestyle and abundant calorie intake, fat deposition has remarkably increased leading to significant consequences such as hypertension, diabetes, coronary heart disease and osteoarthritis (*Padwal & Majumdar, 2007*). Body mass index (BMI) of an individual above 30 kg/m$^2$ or greater tend to possess major comorbidities and the individual is ultimately prone to undertake anti-obesity treatment (*Avenell et al., 2004*). The imbalance between energy expenditure and energy intake is the most significant factor of obesity corresponding to appetite regulation. Anorectic drugs are responsible for the reduction in energy intake which ultimately results in food deprivation leading to weight loss. Activation of anorexigenic hormones is an effective strategy used in managing obesity (*Fu et al., 2016*).

Fat mass and obesity-associated protein (FTO) is involved in food intake and energy expenditure (*Karra et al., 2013*). The expression of FTO is extensively found in the hypothalamus region, namely the arcuate nucleus that helps in regulating energy balance, appetite suppression and energy metabolism (*Frayling et al., 2007*). It plays a crucial role in regulation of homeostasis and controlling energy expenditure by hindering the signals that generate from the brain (*Loos & Yeo, 2014*).

Various therapeutic drugs such as Sibutramine, Rimonabant, Orlistat, Fenfluramine, and Phentermine are employed in treating obesity, but owing to their several concomitant effects other drugs, with the exception of Orlistat, have been withdrawn from the market (*Fu et al., 2016*). The aftermath caused by these drugs were complex and hence an alternative therapy was introduced by using available natural sources that serve as a potential drug target in treating this ailment with minimal or no ill effects.

Bioactive compounds from marine organisms are considered as vital sources in treating obesity (*Hu et al., 2016*). Seaweeds tend to enhance satiety and decrease postprandial levels of glucose and lipids aiding in anti-obesity activity (*Kim et al., 2008*). Brown algae generally possess alginates, phlorotannin, fucoxanthin and fucoidans which serve as potential anti-obesity sources upon exploitation under *in-vivo* conditions. The anti-obesity activity are due to several mechanisms which includes effect on satiety feeling, inhibition of lipid metabolism and inhibition of adipocyte differentiation (*Wan-Loy & Siew-Moi, 2016*). Recently, it was also revealed that red seaweeds are found to possess anti-obesity effects (*Lee et al., 2020*).

Considering the above explanations, the current research deals with the design and development of pursuing an extensive screening of compounds from plant and marine sources through data mining of literature and the database for retrieval of seaweed-based compounds targeting appetite. Following this, the compounds were then analysed through several filters such as ADME and toxicity screening. The protein-ligand interactions were analysed using *in silico* docking approach. The conformational changes and the stability of drug-like compounds were understood by performing molecular dynamic simulation

studies. Finally, the binding energies were calculated for the protein-ligand complex through MM/PBSA analysis to identify the lead compound that regulates appetite.

## METHODOLOGY

### Protein and ligands preparation

FTO protein with 3LFM as corresponding PDB ID was retrieved for this study (*Han et al., 2010*). It possessed chain A that corresponds to the FTO protein and consisted 495 amino acids from *Homo sapiens*. Some missing amino acids were found to be in the loops which showed no interaction at the active site and were excluded from the study. The entire structure had a resolution of 2.50 Å. Prior to docking studies, ligand molecules in the protein structure, non-interacting ions and water molecules were removed using BIOVIA Discovery studio client 2020 (Dassault Systèmes, BIOVIA, Discovery Studio Modelling Environment, Release 2017, San Diego: Dassault Systèmes, 2016).

A library of 100 plant compounds having anti-obesity activity for the FTO protein was retrieved from published literature. The plant sources of these compounds are mentioned in Table S1 and their corresponding structures were downloaded from PubChem database. In addition, 1,110 seaweed compounds retrieved from Seaweed Metabolite Database were provided in Table S2 (*Davis & Vasanthi, 2011*). The 3D structure of all the ligand molecules were generated using Corina and converted to PDB format using an Open Babel module (*O'Boyle et al., 2011*).

### Virtual screening

PyRx v0.8 (*Trott & Olson, 2009*) is a structure-based virtual screening tool with an inbuilt module of Open Babel for ligand processing and AutoDock Vina 1.12 (*Dallakyan & Olson, 2015*) for molecular docking. The 3D structures of the ligands and the protein were loaded in PyRx. The conjugate gradient algorithm was implied to these structures for energy minimization using Universal Force Field (UFF). Energy minimized structures were then converted to PDBQT format. The docking site was defined by 3-methylthymidine (DT) from the same 3LFM structure data. Exhaustiveness modes were set to eight and a total of eight different docking poses were generated and analysed based on binding affinity and interactions with the protein.

### Drug-likeness and oral toxicity prediction

The selected molecules were subjected to ADME predictions using SwissADME (*Daina, Michielin & Zoete, 2017*) to analyse drug likeness property. In an attempt to find optimal bioavailability of the drug, Lipinski's rule of five (RO5) (*Lipinski et al., 2001*) was used as a filter for the virtual screening of compounds. Furthermore, with the aim of achieving significant oral non-toxic drugs, toxicity of the selected molecules was evaluated using ProTox-II server (*Banerjee et al., 2018*). ProTox-II predicts the median lethal dose (LD50) values in mg/kg body weight (*Drwal et al., 2014*) based on the Globally Harmonized System of classification of labelling of chemicals (*Winder, Azzi & Wagner, 2005*).

## Molecular docking

The *in silico* analysis for the selected compounds with 3LFM were performed using AutoDock 4.2.6 with ADT tools of 1.5.6 version (*Morris et al., 1998*). The target protein 3LFM was refined by removal of hetero atoms and water molecules followed by addition of polar hydrogen. Kollaman charges were computed for the receptor and saved in PDBQT format. The calculation of Gasteiger charges was subjected to energy minimized conformations of the ligands. The torsion counts of rotatable bonds were further defined for the ligands and saved in PDBQT format. The Lamarckian Genetic algorithm parameters were applied with a population size of 300, mutation rate of 0.02, energy evaluations of $2.5 \times 10^6$, crossover rate of 0.08 and 100 long GA runs. The docked conformations of each protein and ligand were predicted with the energy values in kcal/mol. The docked output of the protein-ligand complex was rendered using UCSF Chimera v1.15 software (*Pettersen et al., 2004*). Based on the binding affinity, the best docked complexes were visualized using BIOVIA Discovery studio client 2020 (Dassault Systèmes, BIOVIA, Discovery Studio Modelling Environment, Release 2017, San Diego: Dassault Systèmes, 2016). The efficiency of the ligand molecules was assessed using the ligand efficiency calculation as stated by *Hopkins, Groom & Alex (2004)* LE represents average binding energy in relation to number of heavy atoms in a molecule.

## Molecular dynamics (MD) simulation study

The molecular dynamics simulation study was performed using GROMACS 2020.1 (GROningen Machine for Chemical Simulations) software (*Hess et al., 2008*) to explore the structural stability of protein and protein-ligand complexes. The ligand topology and the parameter files were generated using PRODRG server (*Schüttelkopf & van Aalten, 2004*) and protein topology was generated using GROMOS96 53a6 force field (*Oostenbrink et al., 2004*) to produce energy minimized conformation of all protein-ligand complexes. These complexes were solvated using the Extended Simple point charge (SPC-E) water cubic box model (*Wu, Tepper & Voth, 2006*). Protonation state of the residues were determined using PROPKA v2.0 under physiological pH 7.4 (*Bas, Rogers & Jensen, 2008*) and 14 $Na^+$ ions was added to each system to maintain the overall neutrality. Further energy minimization was performed using 50,000 steepest descent steps for each protein-ligand complex. After energy minimization process, each system was equilibrated under NVT (number of particles, volume and temperature) followed by NPT (number of particles, pressure and temperature) for 100 ps time scale of position restrain. Constant temperature was maintained at 300 K using V-rescale temperature coupling method and pressure was maintained at 1 atm with Parrinello-Rahman coupling method (*Martoňák, Laio & Parrinello, 2003*). The particle-mesh ewald (PME) (*Toukmaji et al., 2000*) and Verlet cutoff scheme at 10 kJ mol$^{-1}$ (*Grubmüller et al., 1991*) were used to measure long range electrostatic interactions and van der Waals interactions, respectively. The lincs (Linear Constraint Solver) algorithm was applied for covalent bond constraints (*Hess et al., 1997*). Finally, MD simulation step was carried out to analyse the stability of protein-ligand complexes for 100 ns time scale. The trajectories generated by the simulation were analysed using in-built GROMACS script to evaluate RMSD (root mean square deviation),

radius of gyration (Rg), RMSF (root mean square fluctuation) and hydrogen bonds (H-bond) formation. The graphs were generated and visualized using XMGrace tool.

### Binding free energy calculation

Molecular Mechanics Poisson-Boltzmann Surface Area (MM/PBSA) was used to calculate the binding free energy ($\Delta G$) of protein-ligand complexes using the g_mmpbsav5.12 package in the GROMACS platform (*Kumari, Kumar & Lynn, 2014*). Furthermore, to understand the contribution of each amino acid and its dynamic behaviour, the binding free energy which includes the potential energy (electrostatic and van der Waals' interactions) and free solvation energy (polar and nonpolar solvation energies) of each protein-ligand complex were estimated using the MM/PBSA method. Finally, the stable last 20 ns trajectories of MD simulation for each protein-ligand complexes were selected to compute binding free energy.

## RESULTS AND DISCUSSION

### Virtual screening

The virtual screening of 1,210 compounds was performed using AutoDock Vina in PyRx. Based on the binding energy and their interactions, 443 compounds with binding energy lower than −7.00 kcal/mol were selected for further analysis. The binding energies of 443 compounds are shown in Table S3. The docking protocol was validated by docking 3-methylthymidine ligand into the active site of 3LFM.

### Drug-likeness

Drug-likeness of the screened compounds were evaluated using RO5 parameters. Finally, out of 443 compounds, 369 compounds satisfied Lipinski's properties and exhibited drug-likeness indicating possible permeability and absorption of molecules through biological membranes. RO5 parameters for the 369 compounds are provided in Table S4.

### Toxicity prediction

Acute oral toxicity was evaluated based on ProTox II results. According to *Winder, Azzi & Wagner (2005)*, 30 compounds exhibited non-toxicity with an LD50 value greater than 5,000 mg/kg under class VI. These non-toxic compounds were chosen for the next step of analysis. The toxicity class with LD50 values in mg/kg is listed in Table S5.

### Docking studies

Molecular docking studies were carried out between FTO protein and selected 30 ligands using AutoDock 4.2.6 tools to predict best active compounds. The results obtained proposed that out of 30 compounds docked with the FTO protein, four marine-derived compounds BT012 (24ε-hydroperoxy-6β-hydroxy-24-ethylcholesta-4,-28(29)-dien-3-one), BD064 ((6β,24ε)-24-Hydroperoxy-6-hydroxystigmasta-4,28-dien-3-one), RL442 (10-acetoxyangasiol) and RL074 (2-hydroxyluzofuranone B) exhibited high binding affinity values with more negative docking scores when compared to the reference molecule 3-methylthymidine (DT) as well as plant derived compounds (Table 1).
The affinity of the ligands in the studied was assessed by ligand efficiency (Table 1).

All intermolecular energy interactions and key residues involved in the interactions of 30 compounds are summarized in Table S6.

The docking results obtained pointed out that the reference molecule DT had four conventional hydrogen bonds with Arg96, Tyr106, Asn205 and Arg322 have a binding energy of −6.61 kcal/mol. Nine residues are involved in the hydrophobic interaction and Val228 and His231 residues are involved in Pi alkyl formation (Fig. 1A).

BT012 have binding energy value of −9.12 kcal/mol. Two conventional hydrogen bonds with Tyr106 and Glu234 and a single carbon-hydrogen bond with Thr92 were observed in the binding of BT012 to the active site of 3LFM. In addition, alkyl interaction with Leu109, Val228 residues and Pi alkyl interactions with Pro93, Thr108 and His231 were also observed and are represented in Fig.1B. The conformational binding of BD064 exhibited binding energy of −8.14 kcal/mol which possessed three conventional hydrogen bonds with Tyr106, His 231, Glu 234 and a single carbon-hydrogen with Pro93. Alkyl interactions of Val94 and Val228 were observed in addition to the Pi lone pair of His231 residue (Fig. 1C).

The docked results from compound RL442 had two conventional hydrogen bonds with Lys216 and Ser229 and one carbon-hydrogen bond with Pro93. Hydrophobic interactions formed with Tyr108 and Leu109 had a binding affinity of −8.01 kcal/mol (Fig. 1D). The interactions of the compound RL074 with the active site of 3LFM is depicted in Fig. 1E. The conformational binding exhibited binding energy of −7.5 kcal/mol. The obtained docking results pointed out that formation of conventional hydrogen bonds with Asp233, Glu234 and carbon-hydrogen bonds with Pro93 were important residues to maintain energy conformations. Alkyl interactions of Val94 and Val228 residues were formed, in addition to this Pi lone pair of His231 residue was noted (Fig. 1E).

The four selected marine derived compounds such as BT012 from *Turbinaria conoides* (*Sheu et al., 1999*), BD064 from *Dictyota bartayresiana* (*Rao et al., 1994*), RL442 from *Laurencia sp* (*Vairappan et al., 2010*) and RL074 from *Laurencia saitoi* (*Sun et al., 2005*) showed a lower docking energy score in comparison with DT as well as with plant derived compounds. Pro93, Val94, Tyr106, Tyr108, Leu109, Glu234, Val228 and His231 residues are found to be in common with the reference molecule. These compounds also exhibited strong interaction with the target protein and may be considered as potential inhibitors of FTO. The docked protein-ligand complexes were eventually used to study the structural, dynamic as well as binding behaviours of ligand interactions with the active site of 3LFM using MD simulation studies.

## Molecular dynamics (MD) simulation

MD simulation trajectories were accomplished to assess the detailed structural and dynamic interaction of the FTO protein along with the FTO-ligand complexes (BT012, BD064, RL442, RL074 and DT) for 100 ns time scale. The RMSD-protein backbone, RMSF analysis, hydrogen bond analysis and Rg scores of complexes during the entire simulation period were analysed.

The RMSD of the backbone atoms were assessed to understand the complex stability of complexes over 100 ns time scale. The RMSD of the trajectories revealed that apo FTO

**Table 1 Binding energy and interactions of ligands docked with FTO protein (3LFM).**

| Compound ID | Compound name | Chemical structure | Binding energy (kcal/mol) | Ligand efficiency (kcal/mol) | Interactions with amino acid residues of 3LFM | | |
|---|---|---|---|---|---|---|---|
| | | | | | Hydrogen bond interactions | Hydrophobic interaction | Pi interactions |
| 3-methylthymidine (DT) | FTO inhibitor |  | −6.61 | 0.37 | Arg96, Tyr106, Asn205, Arg322 | Thr92, Pro93, Val94, Tyr108, Leu203, Ser229, Asp234, Glu234, Thr320 | Val228, His231 |
| BT012 | 24ε-hydroperoxy-6β-hydroxy-24-ethylcholesta-4,-28(29)-dien-3-one |  | −9.12 | 0.30 | Thr92, Tyr106, Glu234 | Pro93, Tyr108, Leu109, Val228, His231 | – |
| BD064 | 6β,24ε)-24-Hydroperoxy-6-hydroxystigmasta-4,28-dien-3-one |  | −8.41 | 0.36 | Pro93, Tyr106, His231, Glu234 | Val94, Val228 | His231 |
| RL442 | 10-acetoxyangasiol |  | −8.01 | 0.31 | Pro93, Lys216, Ser229 | Tyr108, Leu109 | – |
| RL074 | 2-hydroxyluzofuranone B |  | −7.5 | 0.35 | Pro93, Asp233, Glu234 | Val94, Val228 | His231 |

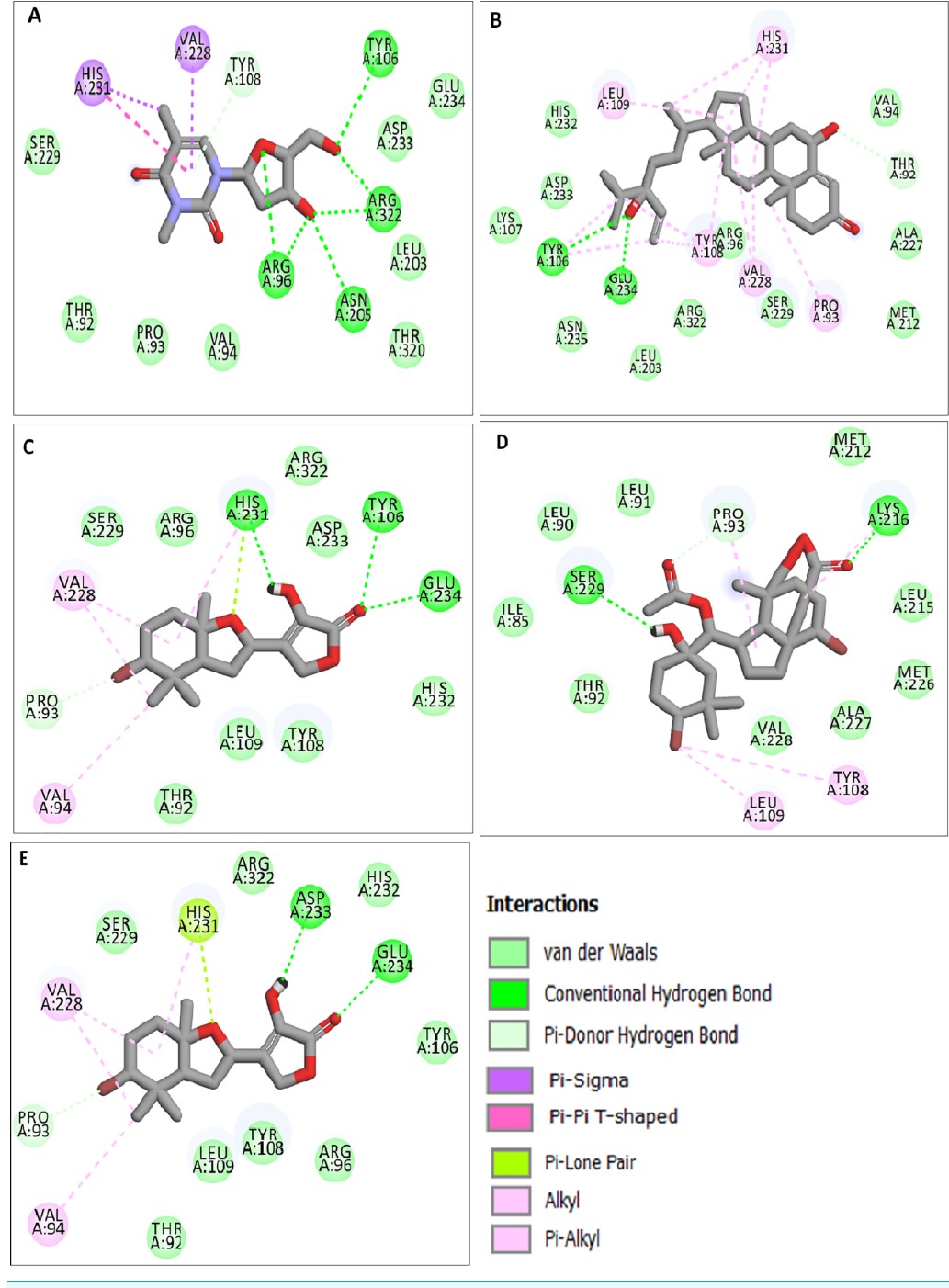

**Figure 1 Docking interactions of ligands in the active site of 3LFM.** 2D representation of (A) DT, (B) BT012, (C) BD064, (D) RL442, (E) RL074.

reached stability after 65 ns whereas all the other FTO-ligand complexes took 20–25 ns to reach mere stability throughout the system. The average RMSD values of FTO protein, FTO-BT012 complex, FTO-BD064 complex, FTO-RL442 complex, FTO-RL074 complex and FTO-DT were found to be 0.39, 0.48, 0.41, 0.34, 0.34 and 0.39 nm,
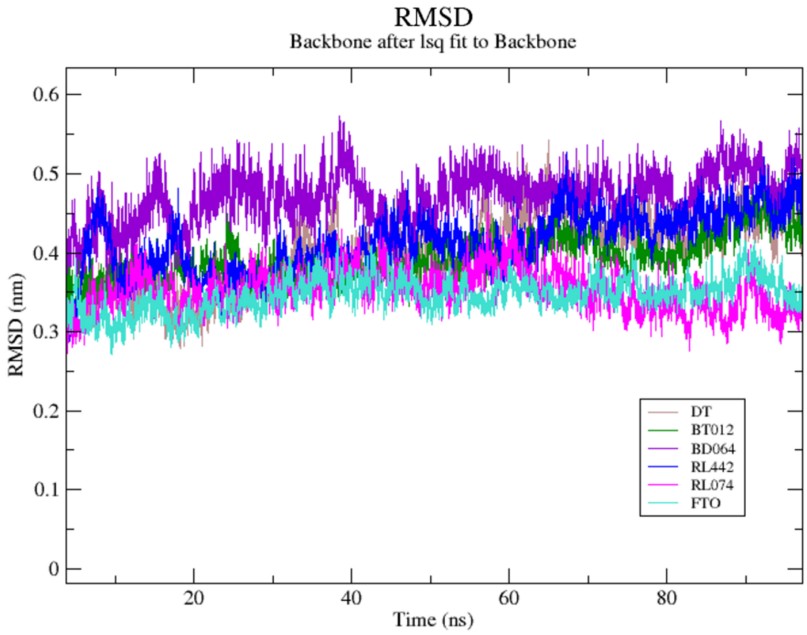

**Figure 2 (A) Superimposed RMSD graph of the backbone atoms in complex for 100 ns simulation time.** (B) Ligand RMSD associated with FTO complexes, overlay and tile conformation. DT (brown), BT012 (green), BD064 (violet), RL442 (blue), RL074 (magenta) and FTO (orange).

respectively. As it was seen that RMSD deviation of protein backbone in selected complexes ranges between 0.2–0.5 nm and does not exceed 0.5 nm which indicates the system to be merely similar and aids in maintaining structural integrity as depicted in Fig. 2. The ligand RMSD plot exhibited stable dynamic profiles in the range of 0.05 to 0.25 nm. The ligand RMSD plot for the ligand complexes are given in Fig. 3. RL074 and RL442 ligands showed stable deviation when compared to reference ligand over 100ns time scale.

The dynamic flexibility and mobility of the residues were computed using RMSF calculations for FTO protein and FTO-ligand complexes. The residues that fluctuate and diverge from the overall structure are represented by peaks in the RMSF plot (Fig. 4). Interestingly, it was seen that RMSF values of all FTO-ligand complexes exhibited an overall lower or similar RMSF value when compared to apo FTO protein during the simulation time of 100 ns. The average RMSF values observed were 0.18, 0.14, 0.18, 0.16, 0.16, and 0.19 nm for FTO-protein, FTO-BT012 complex, FTO-BD064 complex, FTO-RL442 complex, FTO-RL074 complex, and FTO-DTcomplex, respectively.

The compactness and rigidity of the FTO protein and FTO-ligand complexes were investigated using, Rg (Fig. 5). The average Rg value of FTO protein, FTO-BT012 complex, FTO-BD064 complex, FTO-RL442 complex, FTO-RL074 complex and FTO-DT complex were noted to be 2.54, 2.59, 2.54, 2.53, 2.52, and 2.57 nm, respectively. The result from the Rg profile showed that FTO ligand complexes along with the reference remained stable between 2.5 to 2.65 nm throughout the 100 ns time scale. It was observed that

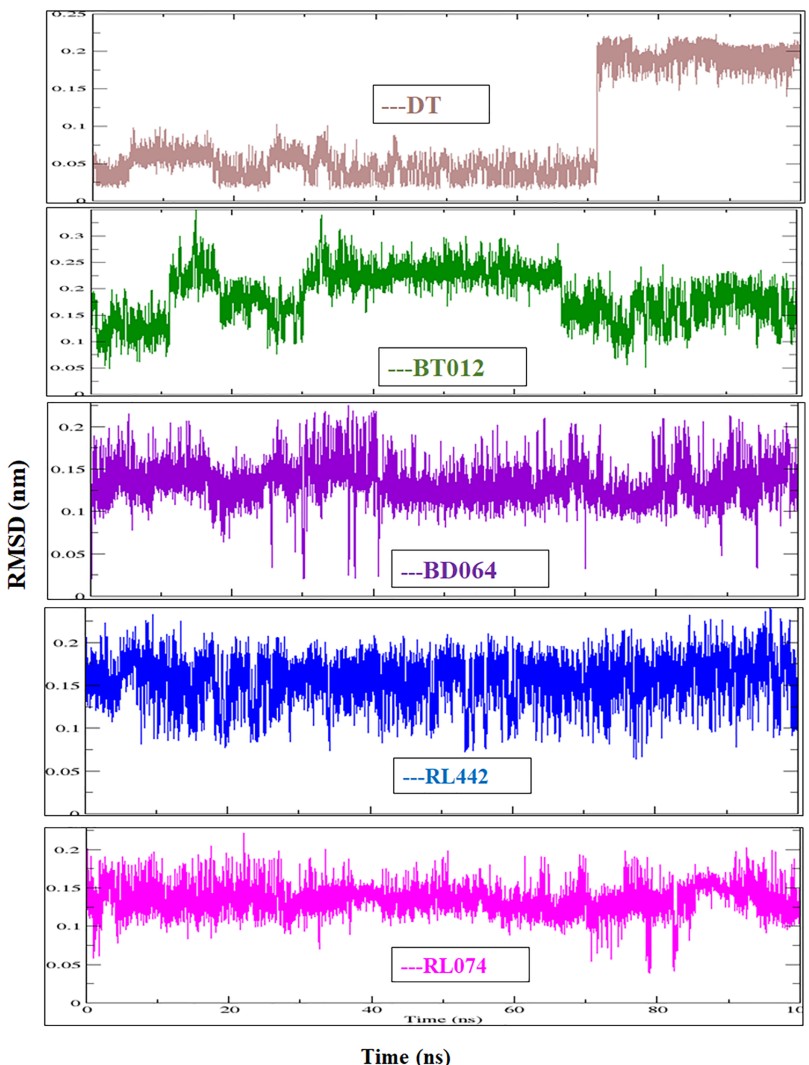

**Figure 3 Ligand RMSD associated with FTO complexes, overlay and tile conformation.** DT (brown), BT012 (green), BD064 (violet), RL442 (blue) and RL074 (magenta).

FTO-RL442, FTO-RL074 complex and FTO-BT012 complex showed lesser gyration and closed confirmation when compared to other complexes.

The formation H-bond is the critical driving force that determines the stability and drug specificity. It was observed that from MD trajectories results of H-bond analysis, fluctuation of H-bonds was between 0 and 4 for FTO-BT012 complex, 0 and 4 for FTO-BD064 complex, 0 and 3 for FTO-RL442 complex, 0 and 5 for FTO–RL074 complex and 0 and 2 for FTO-DT complex (Fig. 6). The average number of H-bonds in brown algal complexes FTO-BT012 and FTO–BD064 were 1.13 and 0.36, respectively, and these complexes had formed a maximum of four bonds. Similarly, the average number of H-bonds in red algal complexes FTO-RL442 and RL074 were 0.20 and 1.73, respectively, in addition FTO-RL074 complex had reached a maximum of five H-bonds.

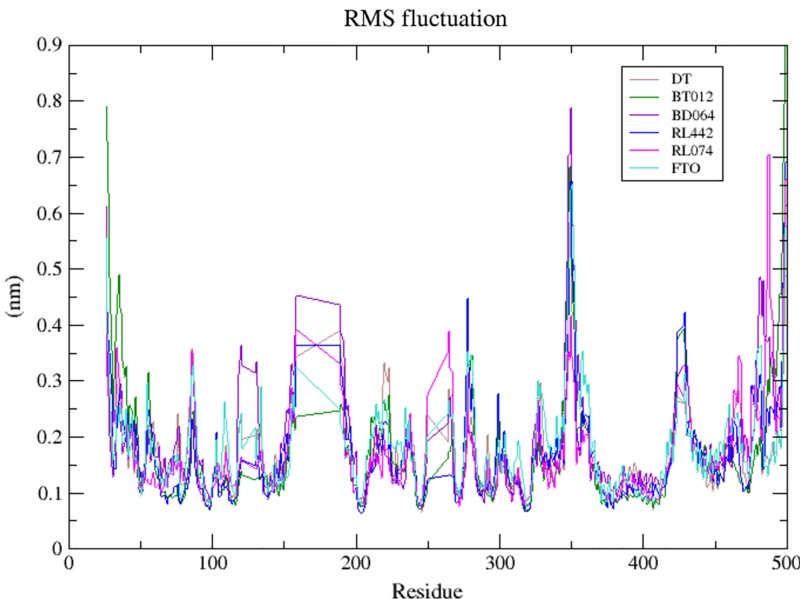

**Figure 4 Superimposed RMSF plots of the backbone atoms of 3LFM in complex for 100 ns MD simulation.** DT (brown), BT012 (green), BD064 (violet), RL442 (blue), RL074 (magenta) and FTO (turquoise).

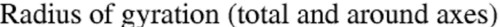

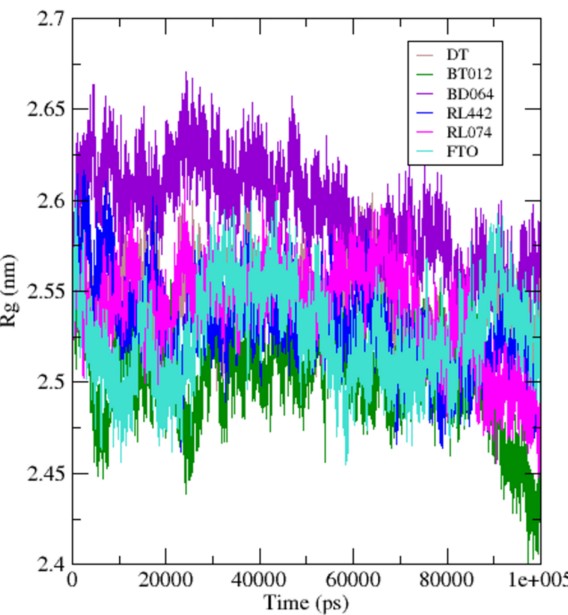

**Figure 5 Radius of gyration of 3LFM in complex with 100 ns time scale.** DT (brown), BT012 (green), BD064 (violet), RL442 (blue), RL074 (magenta) and FTO (turquoise).

Further to verify the post simulation interaction study, three best complexes FTO-BT012, FTO-RL074 and FTO-BT012 were selected along with the control FTO-DT complex (Fig. 7). The final time frame of 100 ns structure of these complexes were dumped

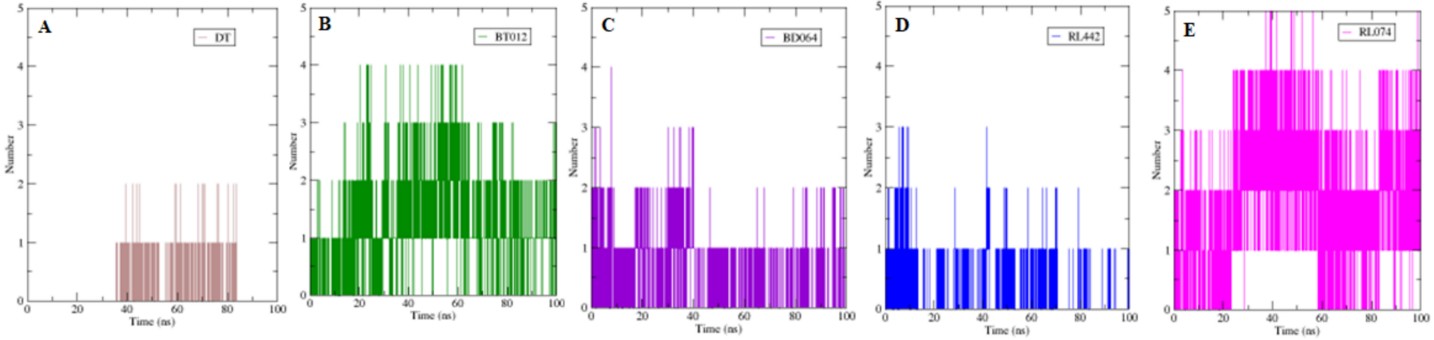

**Figure 6** **The number of hydrogen bonds formed between 3LFM in complex for 100 ns simulation.** (A) DT, (B) BT012, (C) BD064, (D) RL442, (E) RL074.

and analysed based on the H-bond interaction pattern. The post simulation analysis of FTO-BT012 and FTO-RL442 complex reveals that H-bond between Ser229 is present at the binding site of FTO whereas FTO-RL074 reveals H-bond between Thr92 and Arg96. The residues Thr92, Arg96, and Ser229 are found to be present in the active site of FTO inhibitor (DT).

## Binding free energy calculations

To understand the subsequent binding of ligand-protein complexes from dynamic simulation, binding free energy was calculated using the MM-PBSA approach. This method calculates the binding free energy by adding all energy types such as electrostatic energy, polar solvation energy, SASA (Solvent Accessible Surface Area) energy and van der Waals' energy (Table 2).

The binding energies for FTO-BT012 complex, FTO-BD064 complex, FTO-RL442 complex, FTO-RL074 complex, and FTO-DT complexes were subsequently measured for last stable 20 ns MD trajectories. The total binding free energy of the reference molecule FTO-DT complex was −46.24 kJ/mol. Among other complexes, FTO-BT012 exhibited lowest binding energy of −102.06 kJ/mol whereas FTO-BD064 complex, FTO-RL442 complex, and FTO-RL074 complex possessed binding energy of −61.27, −57.40, and −80.24 kJ/mol, respectively. The binding free energy of FTO-ligand complexes was lower than DT. FTO-RL074 possessed lower van der Waals' interaction of −105.94 kJ/mol in comparison with reference molecule that provides stable binding. Also, the FTO-RL074 complex exhibited highest SASA energy value of −13.35 kJ/mol in comparison with DT. In addition to this, the FTO-RL074 complex showed highest electrostatic energy of −86.45 kJ/mol that played a vital role in binding of protein-ligand complex.

The residues that contributed to the overall binding are alone depicted in the Fig. 8 for clear representation of the values. The residues that were majorly involved in energy contribution for all complexes were Val83, Ile85, Leu90, Thr92, Pro93, Leu109, Tyr220 and Met226 (Fig. 8). Interestingly, the abovementioned residues contributed to the active site of protein that plays a major role in maintaining the stability of the FTO-ligand complexes.
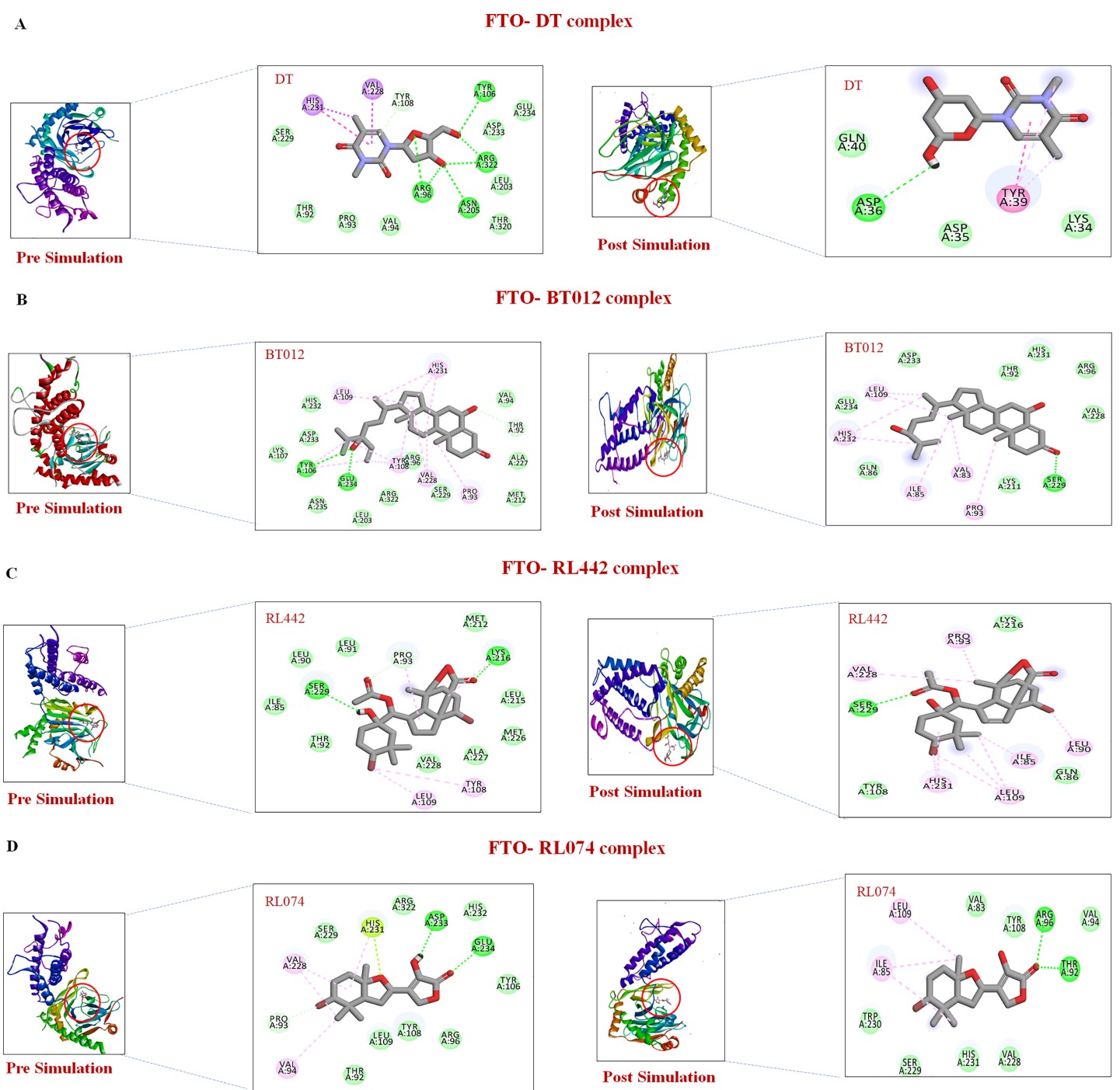

**Figure 7 Pre and post molecular dynamics simulation interaction analysis.** (A) FTO-DT complex. (B) FTO-BT012 complex. (C) FTO-RL442complex. (D) FTO-RL074 complex.

# DISCUSSION

The virtual screening of compounds was sorted based on their estimated binding energy values and their interactions. Owing to this strategy of ranking, 443 compounds with the binding energy lower than −7.00 kcal/mol were selected for further analysis. These

**Table 2 Energy contribution of protein-ligand complexes calculated using MM/PBSA method.**

| Energy parameter (kJ/mol) | FTO-DT complex | FTO-BT012 complex | FTO-BD064 complex | FTO-RL442 complex | FTO-RL074 complex |
|---|---|---|---|---|---|
| Binding energy | −46.24 +/− 84.71 | −102.06 +/− 18.95 | −61.27 +/− 11.73 | −57.40 +/− 18.17 | −80.24 +/− 13.50 |
| Electrostatic energy | −11.54 +/− 12.74 | −35.09 +/− 24.19 | −52.44 +/− 13.78 | −71.45 +/− 21.34 | −86.31 +/− 15.44 |
| Polar solvation energy | 61.62 +/− 43.56 | 50.41 +/− 31.23 | 97.22 +/− 22.87 | 126.14 +/− 31.51 | 122.35 +/− 18.25 |
| SASA energy | −9.01 +/− 9.94 | −12.75 +/− 1.48 | −10.69 +/− 1.15 | −12.35 +/− 1.35 | −13.33 +/− 1.12 |
| Van der Waal energy | −87.30 +/− 94.25 | −104.63 +/− 12.03 | −95.35 +/− 10.56 | −98.73 +/− 11.52 | −105.94 +/− 11.91 |

compounds were refined for Lipinki's property. In general, it states that orally active drugs have no more than one violation of the following parameters: MW ≤ 500, number of hydrogen bond donors ≤5 and hydrogen bond acceptor ≤10 and MlogP ≤ 4.15. A total of 369 compounds that satisfied the above criteria were selected for toxicity prediction. According to *Winder, Azzi & Wagner (2005)*, 30 compounds possessed an LD50 value greater than 5,000 mg/kg under class VI and exhibited non-toxic properties. These non-toxic compounds were chosen for the next step of molecular docking analysis.

Molecular docking studies were carried out between FTO protein and the selected 30 ligands using AutoDock 4.2.6 tools to predict best active compounds. Based on the binding energy and the residues involved in the intermolecular energy interactions, top four compounds BT012, BD064, RL442 and RL074 were selected. These compounds also exhibited strong interaction with the target protein and may be considered as potential inhibitors of FTO. BT012 (24ε-hydroperoxy-6β-hydroxy-24-ethylcholesta-4,-28(29)-dien-3-one), an oxygenated fucosterol derived from brown alga *Turbinaria conoides* exhibited higher binding affinity. Red algal compounds such as RL442 and Rl074 which belonged to the class of Diterpenes and Sesquiterpenes, respectively also exhibited a good docking score and interaction when compared to the reference molecule. The molecular docking results were in accordance with the anti-obesity effects of red seaweed reported by *Lee et al. (2020)*. Overall, the molecular docking results revealed that screened compounds as well as the reference molecule had hydrogen and hydrophobic bonding interactions with the same residues *i.e.*, Pro93, Val94, Tyr106, Tyr108, Leu109, Glu234, Val228 and His231 that are found to be in active site residues of the target protein. The affinity of the ligand molecules in this study was compared and it was observed that the reference molecule DT exhibited highest ligand efficiency. The proposed ligand efficiency score for a ligand molecule should be more than 0.3 kcal/mol. The ligands efficiency scores of top ranked ligands from molecular docking studies vary between 0.30 to 0.37 kcal/mol, thus these ligand molecules can be used a drug (*Schultes et al., 2010*).

Molecular dynamics analysis of the four top ranked complexes was carried out to identify the lead molecule for FTO inhibitor. Interestingly, from RMSD results it was noted that after a 40 ns period, all complexes attained equilibrium and remained stable throughout the entire simulation. Small fluctuation and constant stability after the initial period of fluctuation implied that the system folded to be more stable than the initial structure. The red algal complexes FTO-RL442 and FTO-RL074 were found to exhibit

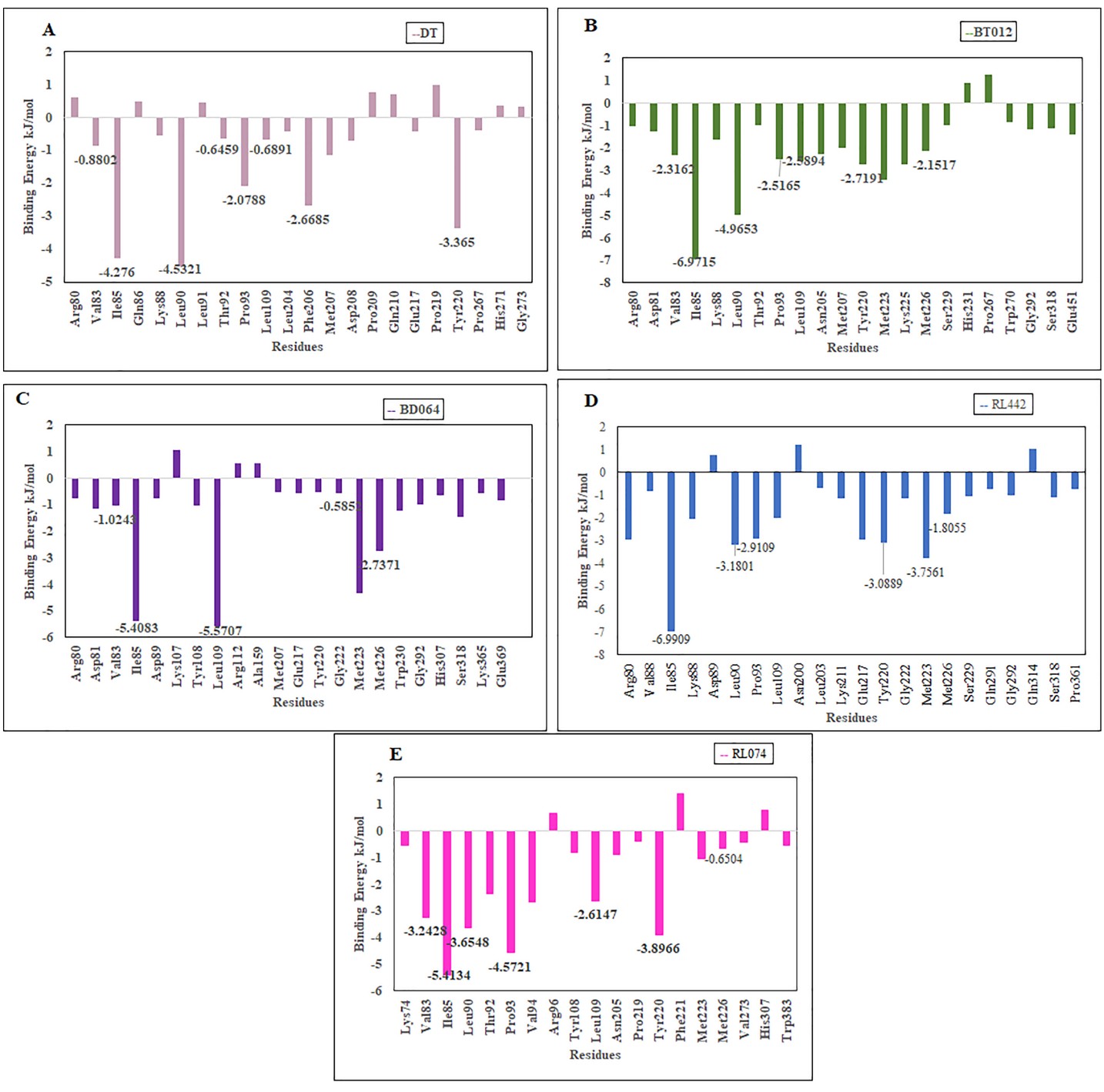

**Figure 8 Binding free energy contribution of each residue of 3LFM in complex.** (A) DT, (B) BT012, (C) BD064, (D) RL442, (E) RL074.

lowest mean RMSD value when compared to the native protein as well as the reference molecule, indicating stable binding of complexes with FTO protein throughout the system. Although RL074 complex exhibited lesser binding energy value, RMSD for FTO-RL074 complex was found to be stable when compared to other complexes. Similarly, the ligand

RMSD of RL074 showed stable fluctuation throughout 100 ns simulation. In addition, the brown algal compounds exhibited smaller deviations when compared to the native FTO protein. The RMSD results showed that upon binding of bioactive compounds, no significant or conformational changes were observed in the FTO structure. Furthermore, the RMSD profile suggested that the last 20 ns stable period was preferable for the next step of structural and dynamics analyses. The ligand RMSD plot showed greater deviation for reference molecule when compared to other ligands.

The dynamic flexibility and mobility of the residues were computed using RMSF calculations. Higher the RMSF value lesser will be the stability whereas lower RMSF values represent good stability of the system. The active site residues such as Lys216, Pro219, Tyr220, and Met297 had fluctuations less than 0.07 nm and were found to be approximately stable during the simulation period. In contrast, the RMSF analysis suggested that higher fluctuation residues were found to be away from the active site.

The compactness and rigidity of complexes were investigated using Rg. Lower Rg value measures the compactness and proper folding stability of the protein structure. This lower $R_g$ values of FTO-BT012 complex, FTO-RL074 complex, can form a narrow cavity (more constraints) at the binding region thereby reducing the ligand dynamism seen in Fig. 5. Initially, little fluctuations were observed but after the 30 ns time period, the Rg of the backbone atoms was reduced for all ligand complexes during simulation. From the Rg result analysis it was noted that all FTO-ligand complexes exhibited more compactness than native protein maintaining stability of the structure throughout the simulation. The higher Rg, that was seen in FTO-BD064 complex can form a broader cavity (less constraint) at the binding site and hence observed relatively high dynamism in BD064.

The number of H-bonds formed had a prominent effect on binding of protein-ligand and it contributes to the overall binding stability of the system during 100 ns simulation time. Apparently, the reference molecule had a maximum of two hydrogen bonds with an average of 1.12. The results suggested that hydrogen bond formation between FTO-RL074 complex computed the prominent stability of the simulation during 100 ns time scale.

Binding free energies of the complexes was calculated using the MM/PBSA method. All the complexes contributed negative binding energy, electrostatic interactions, non-polar solvation energy and van der Waals interactions, while polar solvation energy refined positive energy binding. The binding free energy of FTO-ligand complexes that was lower than DT provides stable binding pattern. Binding of FTO-RL074 was more appreciable due to lower van der Waals' interaction of −105.94 kJ/mol in comparison with the reference molecule that provides stable binding. In addition to this, the FTO-RL074 complex exhibited the highest SASA energy value of −13.35 kJ/mol in comparison with DT as most of the ligand parts were buried into the hydrophobic pocket region. Furthermore, the FTO-RL074 complex showed highest electrostatic energy of −86.45 kJ/mol that played a vital role in binding of the protein-ligand complex. The highest polar solvation energy of 126.14 kJ/mol was exhibited by FTO-RL442 complex. The residue-wise interaction plot suggested that only few residues had positive binding energy, while the

other residues had negative binding energy. Furthermore, the stability of the ligand-protein complex was maintained by the residues with the negative binding energy.

Also, the results of molecular docking and dynamics suggested that the binding site was contributed by hydrophobic residues. Similarly, from MM/PBSA calculation, it can be concluded that the hydrophobic region contributes to the binding pocket, where van der Waals' energy ($\Delta_{Evdw}$) was found to be greater than the electrostatic energy ($\Delta_{Eele}$). Thus, from the binding free energy calculations and molecular docking studies it was suggested that the FTO-BT012 complex had better stability when compared to other complexes and DT.

Prior studies have reported that for drug research development, algal sources were found to possess novel metabolites (*Davis & Vasanthi, 2015*). Recent advancements also highlighted that fucosterol belongs to the class of brown seaweeds and were found to exhibit several therapeutic activities such as antihistaminic, antihyperlipidemic, antidiabetic, anticancer, antiadipogenic, antioxidant, antifungal, antihistaminic, anticholinergic, antiadipogenic, blood cholesterol reduction, and anti-osteoporotic (*Abdul et al., 2016*). Furthermore, the tetraprenyltoluquinols class of compounds isolated from brown algal species *Cystophora fibrosa* showed anti-obesity activity (*Prabhakar et al., 2022*). Previous studies reported that red algal species were found to possess anti-obesity effects (*Lu et al., 2020*; *Yang et al., 2019*). Thus, three seaweed compounds BT012 (24ε-hydroperoxy-6β-hydroxy-24-ethylcholesta-4,-28(29)-dien-3-one), derived from brown alga, RL074 (2-hydroxyluzofuranone B) and RL442 (10-acetoxyangasiol) from marine red alga exhibited good stability when compared to other complexes and the reference FTO inhibitor (DT). These three seaweed compounds may act as a potential FTO inhibitor.

## CONCLUSION

The desire to eat is generally driven by genetic, sociocultural, and emotional factors. The necessity to suppress appetite is increasing notably due to the ill effects caused by fat accumulation in the body. With these factors in mind, in depth virtual screening of plant- and marine-based compounds were analyzed using an *in silico* approach where a total of 1,210 compounds were explored as potential natural drug candidates to target against the FTO protein. In addition to this, further filters like drug-likeness and toxicity prediction of compounds revealed potent therapeutic activity. Molecular docking studies screened top four hits with highest binding affinity towards the active site of the FTO protein. Furthermore, MD simulation studies of FTO-BT012 complex, FTO-BD064 complex, FTO-RL442 complex, FTO-RL074 complex, and FTO-DT complex computed the stability of these complexes through evaluation of RMSD, RMSF, Rg scores and H-bond analysis. Finally, MM/PBSA calculation for the screened compounds leads to identification of potential lead targets based on binding free energy values. Thus, the results suggested that oxygenated fucosterol compound BT012 from brown alga *Turbinaria conoides*, Sesquiterpene compound RL074 (2-hydroxyluzofuranone B) and diterpene compound RL442 (10-acetoxyangasiol) from red alga *Laurencia sp*. could be potent sources of anti-obesity compound by inhibiting the FTO protein.

### Funding

This work was supported by the Founder-Chancellor Shri. N. P. V. Ramasamy Udayar Research Fellowship (U02B190575), and the Sri Ramachandra Institute of Higher Education and Research. The funders had no role in study design, data collection and analysis, decision to publish, or preparation of the manuscript.

### Grant Disclosures

The following grant information was disclosed by the authors:
Founder-Chancellor Shri. N. P. V. Ramasamy Udayar Research Fellowship: U02B190575.
Sri Ramachandra Institute of Higher Education and Research.

### Competing Interests

The authors declare that they have no competing interests.

### Author Contributions

- Lavanya Prabhakar performed the experiments, analyzed the data, prepared figures and/ or tables, and approved the final draft.
- Dicky John Davis G conceived and designed the experiments, analyzed the data, authored or reviewed drafts of the article, and approved the final draft.

### Data Availability

   The raw data is available in the Supplemental Files. The seaweed compounds are also available at the seaweed metabolite database: http://swmd.co.in/download.php.

### Supplemental Information

Supplemental information for this article can be found online at http://dx.doi.org/10.7717/peerj.14256#supplemental-information.

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
