# Peer review of "Computational study of potential inhibitors for fat mass and obesity-associated protein from seaweed and plant compounds"

_PeerJ, doi:10.7717/peerj.14256_

## Round 0.1 · original submission · Major Revisions

Reviewers have raised some serious concerns and shortcomings in the study. MAJOR revision is suggested, which requires substantial and thorough revision to appreciate the quality of the manuscript. Therefore, authors are requested to revise their manuscript in light of reviewers comments. Furthermore, reviewer has suggested that you cite specific references. You are welcome to add them, if you believe they are directly relevant. However, you are not required to include these citations, and if you do not include them, this will not influence my decision.

·

Basic reporting

no comment

Experimental design

no comment

Validity of the findings

no comment

Additional comments

I would like to congratulate the authors for their study in this important field of the treatment of obesity. All efforts are welcome with the intention of indicating good solutions to this hard health problem. I have listed below some general and specific observations, comments, and suggestions.
1-You stated in the introduction section that you are looking for the lead compound that regulates appetite, and the results revealed that three compounds may act as anti-obesity inhibitors. Which of them is the most potent inhibitor?
2- Authors must validate the docking procedure with 3LFM by re-docking the cocrystal ligand and mention how well the bioactive conformation was reproduced by adding the RMSD value.
3. In your opinion, are hydrogen bonds at the origin of the mechanism of action? Or is it a synergy between all the interactions? Cite some works in the bibliography. Below are some references to go with (El Khatabi et al., 2020, 2021, 2022).
El Khatabi, K., Aanouz, I., El-mernissi, R., Khaldan, A., Ajana, M. A., Bouachrine, M., & Lakhlifi, T. (2020). 3D-QSAR and Molecular Docking Studies of p-Aminobenzoic Acid Derivatives to Explore the Features Requirements of Alzheimer Inhibitors. Orbital: The Electronic Journal of Chemistry, 12(4), 172–181. https://doi.org/10.17807/orbital.v12i4.1467
El Khatabi, K., El-Mernissi, R., Aanouz, I., Ajana, M. A., Lakhlifi, T., Khan, A., Wei, D.-Q., & Bouachrine, M. (2021). Identification of novel acetylcholinesterase inhibitors through 3D-QSAR, molecular docking, and molecular dynamics simulation targeting Alzheimer’s disease. Journal of Molecular Modeling, 27(10), 302. https://doi.org/10.1007/s00894-021-04928-5
El Khatabi, K., El-mernissi, R., Aanouz, I., Ajana, M. A., Lakhlifi, T., Shahinozzaman, M., & Bouachrine, M. (2022). Benzimidazole Derivatives in Identifying Novel Acetylcholinesterase Inhibitors: A Combination of 3D-QSAR, Docking and Molecular Dynamics Simulation. Physical Chemistry Research, 10(2), 237–249. https://doi.org/10.22036/pcr.2021.277306.1895
4-Authors can also mention the limitations of the present study in the conclusion section.
5-Typing and grammatical errors should be checked thoroughly.

·

Basic reporting

- The authors use an acceptable level of language in my opinion.
- They provided sufficient references for the statements, especially in introduction and discussion.
- All the figures except for Figure 4 is professional looking. For this figure which is the figure for RMSF, I recommend that x-axis display the residue names instead of residue numbers. I suspect they have missing residues in a loop region in the protein structure as there is not a smooth change from residue numbers 150 to 200 which could be due to missing data points in this region.
- The authors provided raw data file but it is missing the list for 1110 seaweed compounds retrieved from Seaweed Metabolite Database. They should provide a list with IUPAC names of this metabolites and preferably list of SMILES codes for each compound could be provided.
- The manuscript I believe is self-contained and explains the hypothesis of authors with relevant results.

Experimental design

- The manuscript is about an original primary research consistent with the aims and scope of the journal.
- Authors state the research question clearly and mention how their work will be helpful fill the knowledge gap.
- However, some methodology part requires more detailed explanation. Specifically, authors, do not mention how they prepare the protein structure? Did they add missing loop regions, side chain atoms etc? This is important as from RMSF graph it seems like there are missing residues in the structure? How did they deal with that, did they just add the resolved residues together, ignoring the missing residues or another way?
- Also methodology section toxicity prediction part should be more detailed. Briefly mentioning Protox II web server, what kind of algorithm it uses and how the predicted output should be evaluated or considered would be appropriate for more general audiences.

Validity of the findings

- Authors provide all underlying data in their research and provide the results in detail with supporting tables.
- The conclusions are well linked though I find the discussion part more like repeat of results part.
- Related to their results and discussion, they compare the docking scores as well as MM/PBSA scored of suggested compounds with reference compound DT but they do not note that reference compound is actually smaller with less heavy atoms. This could lead to reference compound having less favorable scores in both cases but this effect could be due to the size differences in compounds. Hence it will be good to have ligand efficiency scores calculated and given in Table 1 as well as discussed in appropriate sections.

Additional comments

- Some sentences could be written in a better way as they are difficult to understand. Specificity the ones mentioned below
o Sentence between lines 43-45, two parts connected with “and” are not related and it is not clear who are prone to undertake anti-obesity treatment.
o Sentence between 166-167 should be checked again as I think there are mistakes. The PDB ID mentioned 1LPB is not FTO protein and I think written as a mistake. Also, “validated docking 3 the co-crystalized ligand MUP” part of the sentence is not understood.
o Change “least docking score” to “more negative docking scores” in line 186.
o Change “possessing” to “have” and in fact I don’t think “posses” should be used when mentioning the scores of compounds.
o In line 226, it would be better to refer FTO as apo FTO or Apo form of FTO. The same is valid for line 241, instead of “native”, “apo” could again be used.
o In line 307, there should be space after the decimal between 3LFM.BT012.

Reviewer 3 ·

Basic reporting

• The abstract is lengthy, it should be kept to a minimum.
• Introduction need significant revision.

Experimental design

• How is the active site determined in the docking section? The range is not specified in this part. If it is blind docking, it should be mentioned over there.

Validity of the findings

• Figure 1d is not clear. Authors should give a diagram that is easy to understand.

Additional comments

• The conclusion section should also be revised accordingly.
• The manuscript is loaded with typos and grammatical errors which needs to be fixed.

Annotated reviews are not available for download in order to protect the identity of reviewers who chose to remain anonymous.

---

## Round 0.2 · Minor Revisions

Manuscript is significantly improved by the authors. However, there are still some concerns raised by the reviewers. Please address these concerns and resubmit accordingly.

·

Basic reporting

I have read the revised version of the manuscript “Computational study of potential inhibitors for fat mass and obesity-associated protein from seaweed and plant compounds”. Overall, all the comments have been addressed and the quality of the manuscript has been improved.

Experimental design

I have read the revised version of the manuscript “Computational study of potential inhibitors for fat mass and obesity-associated protein from seaweed and plant compounds”. Overall, all the comments have been addressed and the quality of the manuscript has been improved.

Validity of the findings

I have read the revised version of the manuscript “Computational study of potential inhibitors for fat mass and obesity-associated protein from seaweed and plant compounds”. Overall, all the comments have been addressed and the quality of the manuscript has been improved.

·

Basic reporting

While the authors did answer most of the questions adequately, I believe there should be more explanation about missing loops and protein preparation part. They mentioned that "missing amino acids were found to be in the loops which should no interaction at the active site and were excluded from the study." However, they again did not mention if the missing part are left as it is or they were add these last resolved residues together to have the structure complete. Also, they should mention how pronation state of residues are determined.

Figure 4 could still need to be updated at least to have peak values should be shown, i.e. y-axis limit should be increased to 0.8 or 1 nm maybe.

Other suggested corrections were revised adequately.

Experimental design

no comment

Validity of the findings

no comment

Reviewer 4 ·

Basic reporting

Proper citations have not been made. The background introduction needs to be exhaustive and more clear on the main goals of the paper. The computational aspect to the study has been thorough but molecular docking and MD simulation methodology needs to be elaborated upon.

Experimental design

The authors have analyzed ~443 compounds through computational experiments for the drug likeness and binding affinity to FTO. It was not properly mentioned what is the source of these 442 compound set. No exact references provided. Also authors chose the PDB ID 3LFM for docking and MD simulations, what was the reason for the choice? Is this the only protein structure available in the RCSB? A quick search on Uniprot shows Q9C0B1 · FTO_HUMAN has about 30+ crystal structures. The full length Alfafold model is also deposited. Why was this particular structure chosen needs to be clarified.
Molecular docking is ambiguous and has high false positives... Was any benchmarking done on Autodock Vina? What is the enrichment obtained? Not clear as to why authors would submit such as a half done work.
What was the basis to choose 4-5 complexes for MD runs? Is 100ns enough timescale to run and evaluate conformational motions? What was the charge method used for parametrizing the ligands? How was the protonation states predicted is not clear to the reviewer. No proper citations provided also.

Validity of the findings

Quite questionable to say the least!

Reviewer 5 ·

Basic reporting

In this manuscript, authors reported 3 compounds from Seaweeds which can be potential anti-obesity compounds to target FTO using virtual screening of over 1100 compounds. Overall, this is a good paper but needs grammatical errors checked thoroughly.

Experimental design

The experimental design is thorough and flow logically.

Validity of the findings

No Comment

Additional comments

There are minor issues that should be addressed, which I include here below.

General Comment
1). Line 67: If authors can include in brief "the mechanism by the 4 mentioned bio active compounds exert anti-obesity activity".
2). Line 102: What's the criteria for selecting 3-methylthymidine as reference molecule? As FTO is reported to be active against several methylated nucleobases.
3). Figure 2: Color scheme for DT should be Brown and not Red (I think it's typo).

Minor Issues:
Line 84 "which should no interaction" should be "which showed no interaction".
Line 130: "Molecular Docking Simulation Study", I believe that's the next subsection and should be in Bold.
Line 145 Parinnello-Rahman Method (mention relevant reference).
Line 206 "compound RL074 with the active site of 3LFM is depicted in Figure 1" should be Figure 1e.
Line 217 & Line 304 "potential inhibitors of 3LFM" should be "potential inhibitors of FTO" as 3LFM is a PDB ID.
Line 284 "alone depicted in the figure" which figure?.

---

## Round 0.3 · Minor Revisions

Though the manuscript is significantly improved by the authors, there are still some minor concerns raised by the reviewer. Please address these concerns and resubmit accordingly. Furthermore, I suggest authors to thoroughly check the manuscript for English language and related errors within the manuscript before final acceptance.

·

Basic reporting

The authors revised their manuscript and addressed all the questions/recommendations mentioned. I have no further comments.

Experimental design

The authors revised their manuscript and addressed all the questions/recommendations mentioned. I have no further comments.

Validity of the findings

The authors revised their manuscript and addressed all the questions/recommendations mentioned. I have no further comments.

Additional comments

The authors revised their manuscript and addressed all the questions/recommendations mentioned. I have no further comments.

Reviewer 3 ·

Basic reporting

No comments

Experimental design

No comments

Validity of the findings

No comments

Additional comments

Suggested revisions have not been made, there are still numerous grammatical and typographical mistakes (Lipinkin's property) that have not been fixed.
Some software's name are not correctly written.
Manuscript needs to be read thoroughly.

Reviewer 5 ·

Basic reporting

No Comment

Experimental design

No Comment

Validity of the findings

No Comment

Additional comments

The manuscripts has been improved and results are interesting. Experimental design is thorough. All my comments and concerns have been addressed.

---

## Round 0.4 · accepted · Accept

Manuscript is significantly improved by the authors and now can be accepted in its current form.